# Lipid Nanoparticles Outperform Electroporation in Delivering Therapeutic HPV DNA Vaccines

**DOI:** 10.3390/vaccines12060666

**Published:** 2024-06-17

**Authors:** Mingzhu Li, Lei Liu, Xiaoli Li, Jingran Li, Chao Zhao, Yun Zhao, Xiaopeng Zhang, Panpan He, Xiaoyu Wu, Siwen Jiang, Xingxing Wang, Xiujun Zhang, Lihui Wei

**Affiliations:** 1Department of Obstetrics and Gynecology, Peking University People’s Hospital, No. 11 Xizhimen South Street, Beijing 100044, China; mingzhu1815@bjmu.edu.cn (M.L.); lijingran@pkuph.edu.cn (J.L.); 0062034740@bjmu.edu.cn (C.Z.); zhaoyun@pkuph.edu.cn (Y.Z.); 2Aeonvital Institute of Clinical and Translational Immunology (AICTI), Beijing 102600, China; leiliu@aeonvital.com (L.L.); xiaolili@aeonvital.com (X.L.); zhangxiaopeng@aeonvital.com (X.Z.); panhe@aeonvital.com (P.H.); xiaoyuwu@aeonvital.com (X.W.); siwenjiang@aeonvital.com (S.J.); xingwang@aeonvital.com (X.W.)

**Keywords:** therapeutic HPV vaccine, DNA vaccine, lipid nanoparticles (LNPs), electroporation (EP), cell-mediated immunity

## Abstract

Therapeutic HPV vaccines that induce potent HPV-specific cellular immunity and eliminate pre-existing infections remain elusive. Among various candidates under development, those based on DNA constructs are considered promising because of their safety profile, stability, and efficacy. However, the use of electroporation (EP) as a main delivery method for such vaccines is notorious for adverse effects like pain and potentially irreversible muscle damage. Moreover, the requirement for specialized equipment adds to the complexity and cost of clinical applications. As an alternative to EP, lipid nanoparticles (LNPs) that are already commercially available for delivering mRNA and siRNA vaccines are likely to be feasible. Here, we have compared three intramuscular delivery systems in a preclinical setting. In terms of HPV-specific cellular immune responses, mice receiving therapeutic HPV DNA vaccines encapsulated with LNP demonstrated superior outcomes when compared to EP administration, while the naked plasmid vaccine showed negligible responses, as expected. In addition, SM-102 LNP M exhibited the most promising results in delivering candidate DNA vaccines. Thus, LNP proves to be a feasible delivery method in vivo, offering improved immunogenicity over traditional approaches.

## 1. Introduction

Cervical cancer is the fourth most common cancer among women globally, with 604,000 new cases and 342,000 deaths worldwide [1]. Approximately 90% of cervical cancer is caused by human papillomavirus (HPV), and 70% is caused by high-risk genotypes like HPV-16 and 18 [2,3]. HPV prophylactic vaccine produces neutralizing antibodies targeting the capsid protein L1, effectively preventing HPV infection and offering substantial protection for unexposed women. However, it lacks therapeutic effect for those already infected with HPV [2,4]. Therefore, the development of the therapeutic HPV vaccine becomes imperative. Despite the absence of such a vaccine on the market, efforts are underway. Candidate therapeutic HPV vaccines primarily target the early proteins E6 and E7 of HPV16 and HPV18 [5,6]. These vaccines aim to stimulate the production of E6- and E7-specific CD8+T cells, which play a crucial role in eliminating HPV-infected cells [7,8]. Such advancements hold promise for combating HPV-related diseases, including cervical cancer, by leveraging the body’s immune response against the virus’s oncogenic proteins.

DNA vaccines offer several advantages, including ease of production, product stability, and affordability. However, one drawback is their poor immunogenicity with direct IM injection [9]. To address this limitation, candidate therapeutic HPV DNA vaccines, such as VGX3100, are mainly delivered via electroporation (EP). EP increases cell membrane permeability, facilitating entry of the DNA vaccine into cells, thereby enhancing transfection efficiency and enhance immunogenicity [10]. Despite these benefits, EP delivery has its own set of challenges. It can cause muscle contraction, pain at the injection site, irreversible cell damage, and necessitates specialized equipment [11,12,13,14].

Lipid nanoparticles (LNPs) are lipid-based carriers for delivering nucleic acid [15]. There are currently four FDA-approved LNP products, one for delivery of small interfering RNA (siRNA), Patisiran, and three for delivery of message RNA (mRNA): BNT162b2, mRNA-1273, and mRESVIA. Patisiran is used to treat hereditary transthyretin-mediated (hATTR) amyloidosis, BNT162b2 and mRNA-1273 are both SARS-COVID-19 vaccines, and mRESVIA (mRNA-1345) is the respiratory syncytial virus (RSV) vaccine for older adults [16,17,18,19]. Generally, LNPs contain four common components, which are phospholipid, cholesterol, PEGylated lipid, and ionizable lipid. Among these components, ionizable lipids play a pivotal role in safeguarding RNAs and facilitating their cytosolic transport [15]. These ionizable lipids carry a positive charge at acidic pH levels, enabling the condensation of RNAs into LNPs, yet remain neutral at physiological pH to minimize toxicity [20,21]. Importantly, ionizable lipids such as DLin-MC3-DMA, ALC-0315, and SM-102 have been utilized in clinically approved RNA-LNP formulations, including Alnylam’s Patisiran, Pfizer/BioNTech’s BNT162b2, and Moderna’s mRNA-1273 and mRNA-1345, respectively [20,22]. Moderna’s two vaccines have the same LNP formulation [23]. Increasing attention has been directed towards LNP-encapsulated DNA plasmid delivery to improve cell transfection [14,24]. Therefore, our study aims to explore the feasibility of utilizing LNPs as an effective method to deliver therapeutic HPV DNA vaccines in vivo.

## 2. Materials and Methods

### 2.1. Mice

Female C57BL/6JNifdc mice were 49–62 days old at the time of experiments. All mice were kept under specific-pathogen-free (SPF) conditions in an animal facility at K2 ONCOLOGY. All animal protocols were approved by the Institutional Animal Care and Use Committee at K2 ONCOLOGY.

### 2.2. DNA Vaccine

Two therapeutic HPV DNA vaccines, HPV16E6/E7 and HPV18E6/E7, were constructed using E6/E7 of high-risk HPV-16 and HPV-18 as antigen targets. The DNA sequence encoding HPV16 E6/E7 or HPV18 E6/E7 was inserted into the AVL0318 plasmid.

### 2.3. Preparation of Plasmid-Containing LNP

We use three commercial LNPs to deliver HPV DNA vaccine, including DLin-MC3-DMA LNP A, ALC-0315 LNP B, and SM-102 LNP M, which are the LNPs from Alnylam’s Patisiran, Pfizer/BioNTech’s BNT162b2, and Moderna’s mRNA-1273, respectively. Ionizable lipid, cholesterol, DSPC, and DMG-PEG2000 were dissolved in ethanol at molar ratios of 50:38.5:10:1.5 (for DLin-MC3-DMA LNP-A), 46.3:42.7:9.4:1.6 (for ALC-0315 LNP B), and 50:38.5:10:1.5 (for SM-102 LNP-M). The lipid ratios of each LNP that we used were the same as those for commercial LNP materials. Purified AVL0318-HPV16E6/E7 (expressing E6 and E7 protein of HPV16) or AVL0318-HPV16E6/E7 (expressing E6 and E7 protein of HPV18) was dissolved in 10 mM sodium acetate (pH 3) to 0.15 mg/mL. The ethanol solutions and citrate solutions were mixed using a micromixer (INanoP, Micro & Nano (Shanghai) Biologics Co., Ltd., Shanghai, China). LNPs were dialyzed two times using phosphate-buffered saline (PBS; pH 7.4, Solarbio, Beijing, China) and concentrated with 30-kDa Amicon Ultra centrifuge filters (Millipore, Darmstadt, Germany). DNA was extracted from DNA containing LNP or standard DNA plasmid by the ethanol precipitation method, measured by nanodrop, and then normalized by standard DNA plasmid.

### 2.4. Characterization of DNA Containing LNP

We analyzed the physiochemical properties of the plasmid containing LNP, including particle size, polydispersity index (PDI), and encapsulation efficiency. LNP size (Z-average) was measured by Zeta Potential and Particle Size Analyzer (90Plus PALS, Brookhaven instruments corporation, Austin, TX, USA), and PDI was calculated using software from BIC Particle Solutions (Version 3.6.0.7122). The encapsulation efficiency was measured by gel electrophoresis. Briefly, for the total DNA group, plasmids containing LNP were incubated with 2% Triton for 2 min; there was no treatment for the free DNA group. Then, equal volumes of whole DNA group and free DNA group samples were analyzed by agarose gel electrophoresis in TAE buffer with Ultra GelRed (GR501-AA, Vazyme, Nanjing, China). The gel was visualized under UV light (JB-8360, JOBIN, Beijing, China), and the density of DNA bands was analyzed by the area density tool of Gel-Pro Analyzer software (Version 4.0.00. 001 for Windows 98/T/2000). Then, the encapsulation efficiency (EE%) was calculated: EE% V = [1 − Density (free DNA group)/Density (total DNA group)] × 100%.

### 2.5. Immunization with DNA Vaccines

According to preclinical in vivo experiments with DNA plasmids delivered by LNP, the therapeutic dose of 10 μg was selected [24,25]. Female C57BL/6J Nifdc mice were immunized with 10 μg therapeutic vaccines that each contained a plasmid vector inserted with HPV E6 and E7 sequences and resuspended in 40 μL 1 × TE buffer. Two HPV types (16 and 18) were targeted (16E6/E7 and 18E6/E7 plasmids), with the vector alone serving as a negative control. Three immunization methods were (1) direct intramuscular (IM) injection of naked plasmid, (2) IM injection of naked plasmid plus electroporation (EP), and (3) IM injection of LNP-encapsulated plasmid. DNA vaccines were immunized three times at the quadriceps muscle of the right hind limbs. The mice that compared the immune effect of direct IM vs. IM plus EP (Figure 1) or EP vs. LNP (Figure 2) were immunized once a week. Then, we found that the immune effect of once-every-two-weeks administration was significantly enhanced compared to that of once-a-week administration (Appendix A), so mice that compared the immune effects of the different LNPs (Figure 3) were immunized once every two weeks. Vaccinated mice were sacrificed one week after the third dose, and spleen cells were harvested for immunoassays (IFN-γ ELISpot).

### 2.6. Electroporation

Therapeutic HPV DNA vaccines were given intramuscularly, followed by electroporation using the Nepa21 electroporator (NEPA GENE Co., Ltd., Sonidel, Ireland). The transfer pulses were set to 60 V.

### 2.7. Quantification of Cell-Mediated Immune Responses

The immune effect was evaluated by IFN-γ cytokines produced by E6/E7 antigen-specific CD8+ T cells. HPV16 HPV18-E6/E7-specific CD8^+^ T-cell responses were measured by Mouse IFN-γ ELISpot kits (Cellular Technology Limited (CTL), Cleveland, OH, USA) according to the manufacturer’s protocol. Briefly, 2 × 10^5^ fresh splenocytes were plated in complete RPMI1640 medium (Gibco, Waltham, MA, USA) containing 10% FBS (Gibco) and 1% Penicillin-Streptomycin Solution in 96-well ELISpot plates pre-coated with IFN-γ antibody. Then, cells were restimulated with HPV16/HPV18-E6/E7 peptide pools (concentration: peptide number × 2 μg/mL) for 24 h at 37 ℃. Spots were detected by the protocol and counted using the S6 Ultra ImmunoSpot Analyzer (Cellular Technology Limited (CTL), Cleveland, OH, USA). The number of spots was converted into the number of spots per 1 × 10^6^ cells.

### 2.8. Statistical Analysis

Statistical analysis was performed with Graph Prism 6.01 (Graph Prism software) and presented as means ± SEM. Statistically significant differences were shown at four different levels: * *p* < 0.05; ** *p* < 0.01; *** *p* < 0.001; **** *p* < 0.0001, as indicated for each experiment.

## 3. Results

### 3.1. Comparison of Immune Effect between Direct Intramuscular Injection (IM) and IM Plus Electroporation

We constructed the HPV16 E6/E7 and HPV18 E6/E7 DNA plasmid as a therapeutic HPV DNA vaccine (Figure 1A). Then, we detected the immune effects induced by HPV vaccines under different delivery strategies, including (1) direct intramuscular (IM) injection of naked plasmid, (2) IM injection of naked plasmid plus electroporation, and (3) IM injection of LNP-encapsulated plasmid.

Firstly, the immune effect of the therapeutic HPV DNA vaccine was investigated by direct IM injection of naked plasmid. The experiment was as follows: HPV16E6/E7 and empty vector (EV) naked plasmid were administered by IM injection once a week 3 times. A week after the last administration, splenocytes were harvested and restimulated with E6/E7 peptide pools, and then the cellular immune response was evaluated by the secretion of IFN-γ detected by ELISpot (Figure 1B). From Figure 1G,H, the number of spots in the 16E6/E7 group showed an increased trend compared with that in the EV group, regardless of whether the splenocytes were re-stimulated with the HPV16 E6 or E7 peptide pools (E6: 26.0 EV vs. 59.5 16E6/E7 i.m, *p* = 0.3477 (ns); E7: 8.0 EV vs. 241.0 16E6/E7 i.m, *p* = 0.0515 (ns)). From Figure 1I,J, there was no statistical difference in the number of spots between the HPV18E6/E7 group and the EV group (E6: 30.5 EV vs. 35.5 18E6/E7 i.m, *p* = 0.7102 (ns); E7: 20.0 EV vs. 22.0 18E6/E7 i.m, *p* = 0.790 (ns)). The above experiments indicate that the naked HPV DNA vaccine can only induce a very low immune response by direct IM injection.

To further enhance the immune effect of HPV DNA vaccines, we performed electroporation after IM injection of naked HPV18E6/E7 plasmid in mice, with direct IM injection as the control. IFN-γ ELISpot was used to evaluate the immune effect. From Figure 1G,H, the number of spots in the IM plus electroporation group showed an increased trend compared with that in the direct IM injection group (E6: 59.5 16E6/E7 i.m vs. 324.0 16E6/E7 i.m + EP, *p* = 0.0844 (ns); E7: 241.0 16E6/E7 i.m vs. 767.0 16E6/E7 i.m + EP, *p* = 0.093 (ns)). From Figure 1I,J, the number of spots in the IM plus electroporation group was significantly increased compared with that in the direct intramuscular injection group (E6: 35.5 18E6/E7 i.m vs. 506.0 18E6/E7 i.m + EP, *p* = 0.0025 (**); E7: 22.0 18E6/E7 i.m vs. 75.5 18E6/E7 i.m + EP, *p* = 0.0302 (*)). The above experiments indicated that compared with direct IM injection, IM injection plus electroporation significantly enhanced the immune effect of the naked HPV DNA vaccine.

### 3.2. Comparison of Immune Effect between Naked DNA Vaccine Injection Plus Electroporation and LNP Encapsulation Administration

To investigate whether the LNP-encapsulated HPV DNA vaccine can further enhance the immune effect, we administered ALC-0315 LNP B-encapsulated HPV18E6/E7 plasmid and compared it with naked plasmid injection plus electroporation. We immunized mice once a week. One week after the last dose of immunizations, splenocytes were harvested and re-stimulated with HPV18 E6/E7 peptide pools, then the cellular immune response was evaluated by the secretion of IFN-γ detected by ELISpot (Figure 2A). From Figure 2C, the number of spots in the ALC-0315 LNP B encapsulation group was significantly increased compared with that in the electroporation group (HPV18 E6: 385.0 EP vs. 738.1 LNP, *p* = 0.0041 (**)). From Figure 2D, the number of spots in the ALC-0315 LNP B encapsulation group showed an increased trend compared with that in the electroporation group (HPV18 E7: 72.5 EP vs. 150.6 LNP, *p* = 0.1108 (ns)). The above experiments indicate that ALC-0315 LNP B encapsulation administration significantly enhanced the immune effect of the HPV DNA vaccine compared with electroporation administration.

### 3.3. Comparison of the Immune Effect of Three LNPs for Delivering HPV DNA Vaccine

The above experiments proved that the immune effect of ALC-0315 LNP B delivery was better than that of electroporation delivery. To further investigate the immune effect of different LNPs and select the best LNP materials, we compared three commercial LNPs, including DLin-MC3-DMA LNP A, ALC-0315 LNP B, and SM-102 LNP M, to deliver HPV DNA vaccine. The HPV16E6/E7 and HPV18E6/E7 containing DLin-MC3-DMA LNP A, ALC-0315 LNP B, and SM-102 LNP M samples were characterized by particle size, PDI, and encapsulation. The diameter of HPV16E6/E7 and HPV18E6/E7 containing DLin-MC3-DMA LNP A, ALC-0315 LNP B, and SM-102 LNP M was between 66.11 and 98.28 nm, while their PDI ranged from 0.087 to 0.206. All LNP samples possessed over 98% of the encapsulation efficiency (Appendix A). We first detected the HPV16 or HPV18 E7 protein expression in vitro and found that the expression level of HPV16 or HPV18 E7 protein was low in DLin-MC3-DMA LNP A group (1.87% and 0.49%, respectively), higher in ALC-0315 LNP B group (39.1% and 22.4%, respectively) and highest in SM-102 LNP M group (69.0% and 64.2%, respectively) (Appendix A).

The mice in the LNP groups were immunized with HPV16 or HPV18E6/E7 plasmid encapsulated by three types of LNPs, while the mice in the control groups were immunized with naked plasmid without LNP encapsulation once every two weeks. One week after the last dose of immunizations, splenocytes were harvested and re-stimulated with HPV16 or HPV18 E6/E7 peptide pools, then the cellular immune response was evaluated by the secretion of IFN-γ detected by ELISpot (Figure 3A).

Firstly, we investigated whether the three LNP groups all induced an adequate immune response compared with naked DNA groups. From Figure 3F,G, mice were immunized with HPV16 E6/E7 DNA vaccine encapsulated by different LNPs. For re-stimulation with HPV16 E6 peptide pools, the number of spots in SM-102 LNP M but not DLin-MC3-DMA LNP A group was significantly increased compared with that of the naked DNA group (control group) (HPV16 E6: 14.0 naked vs. 289.5 LNP M, *p* = 0.0071 (**)) (Figure 3F). The number of spots in the ALC-0315 LNP-B group showed an increased trend compared with that of the naked DNA group (HPV16 E6: 14.0 naked vs. 34.5 LNP B, *p* = 0.1271 (ns)) (Figure 3F). For re-stimulation with HPV16 E7 peptide pools, the number of spots of ALC-0315 LNP B and SM-102 LNP M but not DLin-MC3-DMA LNP A was significantly increased compared with that of the naked DNA group (HPV16 E7: 315.5 naked vs. 107.0 LNP A vs. 1053 LNP B vs. 1394 LNP-M, *p* = 0.1362 (ns) (naked vs. LNP A), *p* = 0.0005 (***) (naked vs. LNP B) *p* = 0.0040 (**) (naked vs. LNP M)) (Figure 3G). These data indicate that ALC-0315 LNP B- and SM-102 LNP M- but not DLin-MC3-DMA LNP A-encapsulated HPV16 E6/E7 DNA vaccine induces both HPV 16 E6 and E7 antigen-specific immune responses.

From Figure 3H,I, mice were immunized with HPV18 E6/E7 DNA vaccine encapsulated by different LNPs. For re-stimulation with HPV18 E6 and E7 peptide pools, the number of spots of DLin-MC3-DMA LNP A, ALC-0315 LNP B, and SM-102 LNP M is all significantly increased compared with that of the naked DNA group (HPV18 E6: 174.5 naked DNA vs. 721.0 LNP-A vs. 1441 LNP B vs. 2953 LNP M, *p* = 0.0179 (*) (naked vs. LNP A), *p* = 0.0032 (**) (naked vs. LNP B), *p* = 0.0036 (**) (naked vs. LNP M); HPV18 E7: 19.5 naked vs. 302.5 LNP A vs. 756.0 LNP B vs. 2114 LNP-M, *p* = 0.0026 (naked vs. LNP A), *p* = 0.0024 (naked vs. LNP B), *p* = 0.00227 (naked vs. LNP M)). These data indicate that DLin-MC3-DMA LNP A-, ALC-0315 LNP B-, and SM-102 LNP M-encapsulated HPV18 E6/E7 DNA vaccine all significantly induce HPV18 E6 and E7 antigen-specific immune responses.

The above data proved that ALC-0315 LNP B and SM-102 LNP M groups significantly induced strong HPV16 and 18 E6/E7 antigen-specific CD8+ T cell immune responses, while the LNP A group only induced weak HPV18 but not HPV16 E6/E7 antigen-specific CD8+ T immune responses.

Next, we compared the immune effect in DLin-MC3-DMA LNP A, ALC-0315 LNP B, and SM-102 LNP M groups and chose the best material to deliver the HPV DNA vaccine. For all re-stimulations with peptide pools, the number of spots in the SM-102 LNP-M group is significantly increased compared with that in DLin-MC3-DMA LNP-A groups (Figure 3F–I). For HPV16 E6 peptide pools re-stimulation, the number of spots in the SM-102 LNP-M group is significantly increased compared with that in ALC-0315 LNP-B groups (HPV16 E6: 34.5 LNP--B vs. 289.5 LNP M, *p* = 0.0109 (*)) (Figure 3F). For re-stimulation with HPV16 E7 or HPV 18 E6/E7 peptide pools, the number of spots in the SM-102 LNP-M group showed an increased trend compared with that in the ALC-0315 LNP-B group due to the large variation (HPV16 E7: 1053 LNP B vs. 1394 LNP M, *p* = 0.2549 (ns); HPV18 E6: 1441 LNP B vs. 2953 LNP-M, *p* = 0.0758 (ns); HPV18 E7: 756.0 LNP B vs. 2114 LNP-M, *p* = 0.1130 (ns)) (Figure 3G–I). In summary, the SM-102 LNP M group showed the highest number of spots, the ALC-0315 LNP B group showed the lower number of spots, and the DLin-MC3-DMA LNP A group had the lowest number of spots, and these results showed a positive correlation with the cell transfection efficiency in vitro (Appendix A). These data indicate that the HPV DNA vaccine encapsulated by different LNPs shows different immune effects. In this study, SM-102 LNP M encapsulation for delivering HPV DNA vaccine shows the strongest immune effect.

IFN-γ detected by ELISpot reflects HPV antigen-specific T-cell responses to therapeutic HPV DNA vaccine (Figure 4). From our results, naked HPV DNA vaccine by direct IM injection can only induce a very low immune response. IM injection plus electroporation enhanced the immune effect of the naked HPV DNA vaccine. LNP encapsulation (e.g., ALC-0315 LNP B and SM-102 LNP M) for delivering the HPV DNA vaccine shows strong immune effects (Figure 4). In conclusion, LNP encapsulation is the optimal method for delivering a therapeutic HPV DNA vaccine.

## 4. Discussion

There is currently no therapeutic HPV vaccine available on the market. The most promising candidate in this regard is the DNA vaccine VGX-3100, which has completed two phase III trials. These trials have demonstrated that VGX-3100 has met its primary endpoint, suggesting that it could become the first therapeutic HPV vaccine on the market [26]. VGX-3100 targets the E6/E7 protein of HPV16 and HPV18 and is administered by IM injection via electroporation. This method involves administering the vaccine intramuscularly and then using a specialized electroporation device to enhance cellular uptake by creating electric field pulses, thereby boosting the vaccine’s immune response. Our study also observed that IM injection combined with electroporation can indeed enhance the immune effect of the therapeutic HPV vaccine. However, there are three disadvantages in the actual vaccination of therapeutic HPV vaccine: (1) Electroporation administration may cause muscle contraction or pain, leading to significant discomfort and reduced patient compliance; (2) the requirement for specialized electroporation equipment increases the overall cost of vaccine administration; and (3) electroporation-based drug delivery necessitates specialized training for medical personnel to ensure proper administration. Improper operation could result in severe local cell damage or even fatalities. In summary, the promotion of therapeutic HPV vaccines based on electroporation technology will have certain difficulties, especially in low- and middle-income countries.

Currently, the FDA has approved three mRNA-LNP drugs: BNT162b2, mRNA-1273, and Patisiran [15,27]. BNT162b2 and mRNA-1273 are both mRNA-based vaccines developed for SARS-COVID-19. Amid the COVID-19 pandemic, a considerable number of individuals received these vaccines, with adverse events primarily being transient and mild to moderate, both locally and systemically, and with few serious adverse events reported, comparable to those in the placebo group [16,17]. Patisiran, on the other hand, is a small interfering RNA (siRNA) used in the treatment of hereditary transthyretin-mediated (hATTR) amyloidosis, with common adverse reactions including upper respiratory tract infections and infusion-related reactions [18]. In summary, LNP serves as a very safe encapsulation material for nucleic acid drugs.

Given the limited acceptance of therapeutic HPV DNA vaccine based on electroporation technology, this study mainly explored the immune response of an LNP-encapsulated therapeutic HPV DNA vaccine. Surprisingly, the study found that delivery via ALC-0315 LNP B was superior to electroporation. Furthermore, it was observed that SM-102 LNP M delivery produced better immune effects compared to ALC-0315 LNP B, suggesting that SM-102 LNP M is the most effective material for delivering therapeutic HPV DNA vaccines. All LNPs share similar physiochemical properties, including particle size, polydispersity index (PDI), and encapsulation efficiency. We hypothesize that the divergent immune responses triggered by LNPs stem from variations in their ionizable lipids. LNP A contains the ionizable lipid DLin-MC3-DMA, which possesses two tails and creates a subtle cone shape. Conversely, LNP B and LNP M, containing ALC-0315 and SM-102 as ionizable lipids, respectively, share a similar structure, each comprising four tails and having a more pronounced cone shape, which leads to more effective endosomal escape [28], thereby enhancing antigen expression. Furthermore, SM-102 LNP M exhibits superior performance over ALC-0315 LNP B in terms of intramuscular mRNA delivery and antibody production in mice [29]. In our study, we found that ALC-0315 LNP B and SM-102LNP M vaccines induce stronger immune effects compared with DLin-MC3-DMA LNP A vaccines, and SM-102 LNP M shows better immune effects for delivering DNA vaccine as mRNA delivery compared with ALC-0315 LNP B; these data were consistent with the previous results [28,29].

However, limitations of using LNP remain as follows: the LNP vehicle has the intrinsic adjuvanticity of mRNA-LNP vaccines. The LNP itself could induce a variety of signals, including chemokines and pro-inflammatory cytokines [30]. Therefore, the DNA-LNP vaccine may also have some side effects like mRNA-LNP vaccines, including fever, fatigue, chills, and so on. Besides, the number of DNA plasmid copies and the levels of expressed E6/E7 antigenic proteins were not detected in different tissues and specific cells, so the exact relationship between cellular immunity and antigen under different delivery strategies remains to be further studied in the future.

In conclusion, compared with electroporation delivery, the selection of appropriate LNPs to encapsulate DNA vaccines not only enhances the immune effect but also simplifies and expedites the DNA vaccination process. LNP-encapsulating DNA vaccines have the potential to significantly broaden the vaccine market. In addition to this, the LNP encapsulation delivery method holds valuable insights for the delivery of DNA vaccines targeting other viruses and pathogens.

## Figures and Tables

**Figure 1 vaccines-12-00666-f001:**
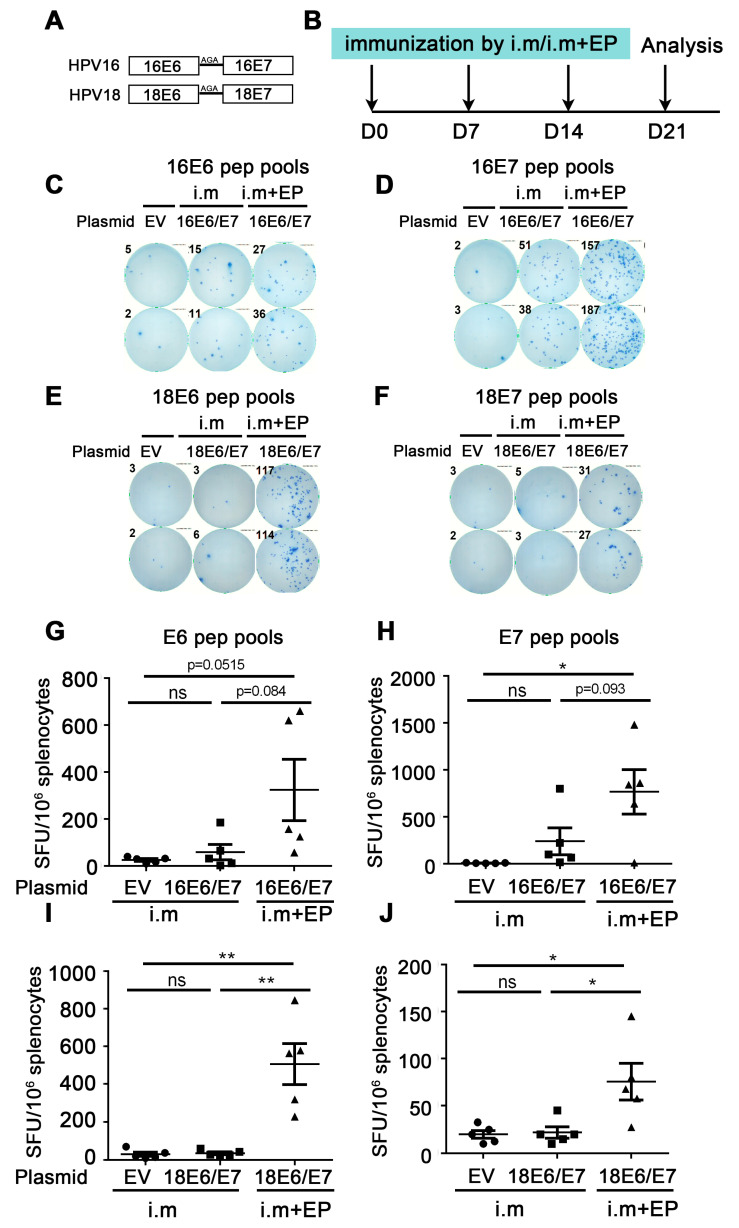
Comparison of immune effect between direct intramuscular injection (IM) and IM plus electroporation. (**A**) Schematic diagram of DNA vaccine sequence design. HPV16 or HPV18 E6 and E7 were fusion expressions and linkers by Ala-Gly-Ala (AGA). (**B**) The vaccine schedule is shown. Mice were injected with empty vector, HPV16, or HPV18 E6/E7 plasmid by direct IM or IM plus electroporation, and the immune effect was evaluated by IFN-γ ELISpot. The administration interval was once a week, three times in total. A week after the last administration, splenocytes were harvested and restimulated with E6/E7 peptide pools, and then the cellular immune response was evaluated by the secretion of IFN-γ detected by ELISpot. (**C**,**D**,**G**,**H**) Splenocytes were restimulated with HPV16 E6 or HPV16 E7 peptide pools, respectively. (**E**,**F**,**I**,**J**) Splenocytes were restimulated with HPV18 E6 or HPV18 E7 peptide pools, respectively. (**C**–**F**) Representative good images. (**G**–**J**) Statistical data. The horizontal coordinates represent different immune conditions; the ordinate represents the number of spots displayed by IFN-γ ELISpot per 106 splenocytes. “i.m” is short for IM injection; “EP” is short for electroporation. Bars represent mean ± SEM of n = 5 independent experiments. ns, not significant; * *p* < 0.05; ** *p* < 0.01.

**Figure 2 vaccines-12-00666-f002:**
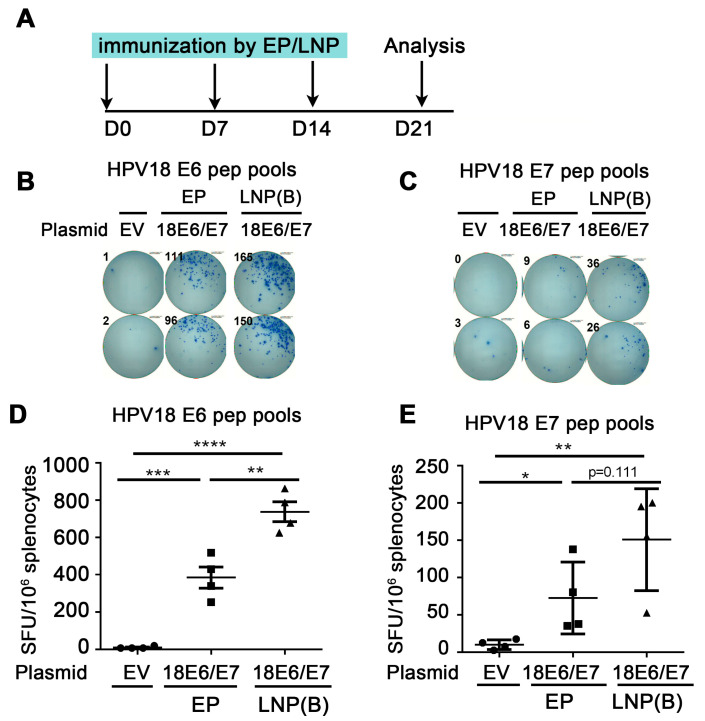
Comparison of immune effect between electroporation and LNP encapsulation administration. (**A**) The vaccine schedule is shown. The HPV18 DNA vaccine was administered to mice by IM injection plus electroporation or ALC-0315 LNP B-encapsulated plasmid IM injection. The administration was once a week, 3 times. A week after the last immunization, the splenocytes were harvested and restimulated by HPV18 E6 (**B**,**D**)/E7 (**C**,**E**) peptide pools. The immune effect was evaluated by IFN-γ detected by the ELISpot experiment. (**B**,**C**) A representative well image. (**D**,**E**) Statistical data. The horizontal coordinates represent different immune conditions. The ordinate represents the number of spots displayed by IFN-γ ELISpot per 10^6^ splenocytes. “EP” is short for electroporation. “LNP” is short for lipid nanoparticle. Bars represent mean ± SEM of n = 4 independent experiments. ns, not significant; * *p* < 0.05; ** *p* < 0.01; *** *p* < 0.001; **** *p* < 0.0001.

**Figure 3 vaccines-12-00666-f003:**
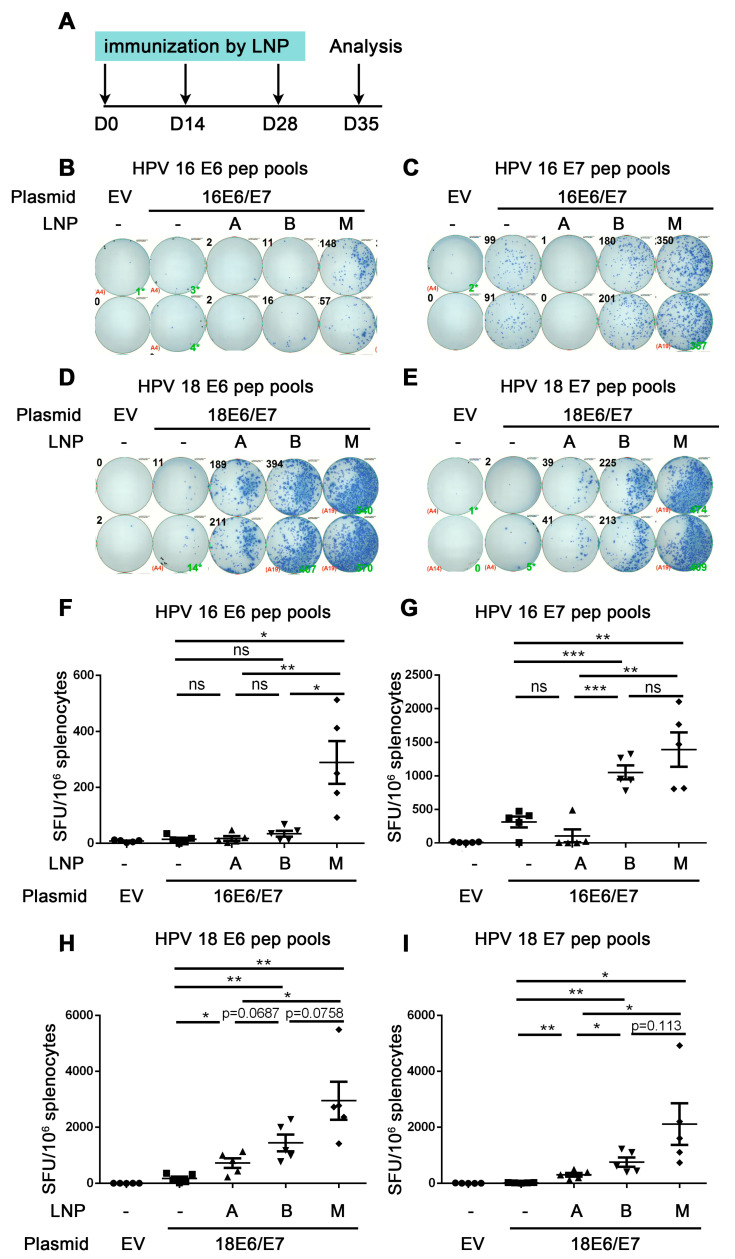
Comparison of the immune effect of three LNPs for delivering HPV DNA vaccine. A. The vaccine schedule is shown. The HPV16 or HPV18 DNA encapsulated by DLin-MC3-DMA LNP (**A**), ALC-0315 LNP B, and SM-102 LNP M was intramuscularly injected into mice. The immunization was once every two weeks, 3 times. A week after the last immunization, the splenocytes were harvested and restimulated by HPV16 E6 (**B**,**F**)/E7 (**C**,**G**) HPV18 E6 (**D**,**H**)/E7 (**E**,**I**) peptide pools. IFN-γ was detected by ELISpot to evaluate the immune effect. (**B**–**E**) shows a representative good image. (**F**–**I**) shows the statistical data. The horizontal coordinates represent different immune conditions. The ordinate represents the number of spots displayed by IFN-γ ELISpot per 10^6^ splenocytes. “LNP” is short for lipid nanoparticle. Bars represent mean ± SEM of n = 5 independent experiments. ns, not significant; * *p* < 0.05; ** *p* < 0.01; *** *p* < 0.001.

**Figure 4 vaccines-12-00666-f004:**
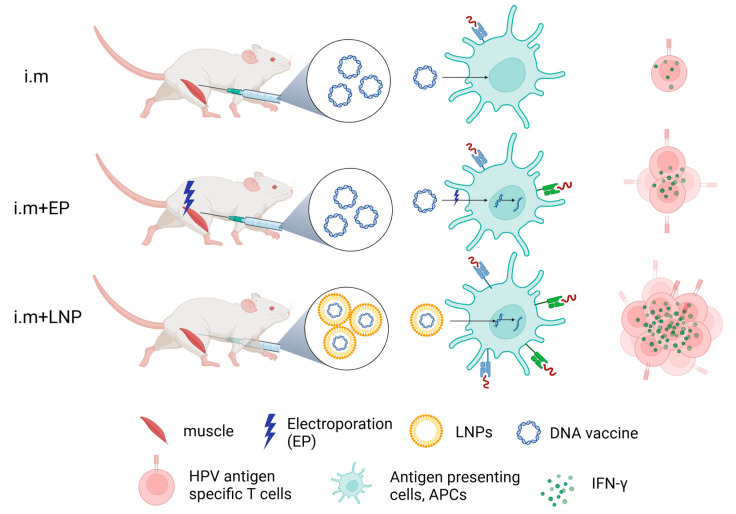
Schematic diagram of immune effects of therapeutic HPV DNA vaccine with different delivery strategies. Created with BioRender.com (DBA BioRender #2961-9728).

## Data Availability

The data can be shared up on request.

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
