# Peer review of "Lipid Nanoparticles Outperform Electroporation in Delivering Therapeutic HPV DNA Vaccines"

_vaccines, 2024, doi:10.3390/vaccines12060666_

Round 1

Reviewer 1 Report

Comments and Suggestions for Authors

The manuscript discusses the use of lipid nanoparticle (LNP) vehicles to deliver a therapeutic DNA vaccine against HPV and compares the LNP vehicle to electroporation. The study question is interesting and the application is significant. However, major comments and questions about the investigation and manuscript remain:

1. The compositions (lipid ratios) of each of the LNPs are different. How were they decided and why was it different for each ionizable lipid? This is a major issue and is another factor to be considered other than the type of ionizable lipid

2. The LNP physiochemical properties must be included. How were the particles characterized? Size, encapsulation, morphology, etc?  The LNPs could have major differences in properties for each ionizable lipid (and composition) that could influence cell internalization and ultimately efficacy. This data is needed.

3. How was the DNA concentration determined post Amicon filtering step of the LNP fabrication?

4. How was the 10 ug dose determined?

5. A vaccination schedule is needed to better understand frequency and type of treatment and points of analysis.

6. The discussion section is missing discussion of the data and why LNP-M is superior to the others. Is it composition? Is it the cationic lipid? Is it the physiochemical properties? What are the limitations of using LNPs especially for DNA vs other cargo? All those critical points are missing. Some of these must also be included in the introduction.

Author Response

  1. The compositions (lipid ratios) of each of the LNPs are different. How were they decided and why was it different for each ionizable lipid? This is a major issue and is another factor to be considered other than the type of ionizable lipid

Reply: Yes, we use three commercial LNPs to delivering HPV DNA vaccine, including DLin-MC3-DMA LNP A, ALC-0315 LNP B and SM-102 LNP M, which are the LNPs from Alnylam’s patisiran, Pfizer/BioNTech’s BNT162b2, Moderna’s mRNA-1273, respectively. Ionizable lipid, cholesterol, DSPC, and DMG-PEG2000 were dissolved in ethanol at a molar ratio of 50:38.5:10:1.5 (for DLin-MC3-DMA LNP-A), 46.3:42.7:9.4:1.6 (for ALC-0315 LNP B), 50:38.5:10:1.5 (for SM-102 LNP-M). The lipid ratios of each LNPs that we used were same as commercial LNP materials.

Yes, the lipid ratios of LNPs are very important, we will change the lipid ratios to get the best delivering results in the future.

  1. The LNP physiochemical properties must be included. How were the particles characterized? Size, encapsulation, morphology, etc?  The LNPs could have major differences in properties for each ionizable lipid (and composition) that could influence cell internalization and ultimately efficacy. This data is needed.

Reply: Thank you for your advice, we have added the data of particle size, PDI and encapsulation efficiency in the figure 2S.

  1. How was the DNA concentration determined post Amicon filtering step of the LNP fabrication?

Reply: DNA was extracted from DNA containing LNP or stander DNA plasmid by ethanol precipitation method and measured by nanodrop, then normalized by stander DNA plasmid. Briefly, 100 μL LNP sample was added to 900 μL absolute alcohol with a vertex of 15 s. Then centrifuge 14000 rpm for 15 min at 4℃ and discard supernatant. Add another 1000 absolute alcohol, centrifuge 14000 rpm for 15 min at 4℃ and discard supernatant. After the ethanol has completely evaporated, the sample were resuspended in 100 μL (equal volume to LNP samples) ultrapure water and measured the DNA concentration by nanodrop. We also do the stander DNA plasmid with the same ethanol precipitation method and calculate the DNA recovery rate. Then calculate the DNA concentration of LNP samples. The above has been supplemented in the method section.

  1. How was the 10 ug dose determined?

Reply: According to preclinical in vivo experiments of DNA plasmid delivered by LNP, 0.5-25μg was administered to each mouse. (Algarni et al., 2022; Guimaraes et al., 2024; Qin et al., 2024; Scalzo et al., 2022; Zhang et al., 2023). The dose for administering in most of articles were 10 μg, so we also chose 10 μg. The above has been supplemented in the method section.

  1. A vaccination schedule is needed to better understand frequency and type of treatment and points of analysis.

Reply: Thank you for your valuable advice. The vaccination schedule is added in the figure 1,2,3.

  1. The discussion section is missing discussion of the data and why LNP-M is superior to the others. Is it composition? Is it the cationic lipid? Is it the physiochemical properties? What are the limitations of using LNPs especially for DNA vs other cargo? All those critical points are missing. Some of these must also be included in the introduction.

Reply: Thank for your advice. These have been discussed in discussion section.

Reviewer 2 Report

Comments and Suggestions for Authors

Thanks to the authors for the manuscript contributing valuable insights into the effective HPV therapeutic vaccines. The authors have provided a study with data that may have implications for future clinical applications.

The following major points are recommended to be revised:

1. Please include a schematic diagram of the DNA vaccine construct used in the experiments.

2. Please verify whether the E6/E7 proteins inserted in the DNA vaccine are expressed in vitro and in vivo.

3. Representative well images for each treatment group should be presented in the ELISpot IFN-γ assay.

4. Given that the manuscript mentions lipid nanoparticles, please add images demonstrating the physicochemical properties of the various LNPs used in the experiments, such as particle size, zeta potential, absorption spectrum, and stability in physiological solutions.

5. Since the manuscript states that “SM-102 (LNP-M) is the most effective material for delivering therapeutic HPV DNA vaccines,” please include data demonstrating the differences in cellular delivery efficiency among the various LNPs and EP used in the experiments. Results should be presented to show the differences in the number of DNA plasmid copies delivered into target tissue regions or specific cells under different delivery strategies, as well as the differences in the levels of expressed E6/E7 antigenic proteins. Of course, methods such as tissue fluorescence distribution or histochemistry are also very acceptable.

Author Response

The following major points are recommended to be revised:

  1. Please include a schematic diagram of the DNA vaccine construct used in the experiments.

Reply: Thank you for your advice, we have added the schematic diagram of the DNA vaccine construct in Figure 1A

  1. Please verify whether the E6/E7 proteins inserted in the DNA vaccine are expressed in vitroand in vivo.

Reply: Yes, we detect the HPV16 and HPV18 E7 protein expression by flow cytometry in vitro. And the data were in Figure S3.

  1. Representative well images for each treatment group should be presented in the ELISpot IFN-γ assay.

Reply: Thank you for your advice. Representative well images were added in the Figure 1,2,3.

  1. Given that the manuscript mentions lipid nanoparticles, please add images demonstrating the physicochemical properties of the various LNPs used in the experiments, such as particle size, zeta potential, absorption spectrum, and stability in physiological solutions. 

Reply: Yes, we have added the data of particle size, PDI and encapsulation efficiency in the Figure 2S. Thank you for your valuable advice, absorption spectrum and stability were not involved in our test, but corresponding detection will be carried out in the next step.

  1. Since the manuscript states that “SM-102 (LNP-M) is the most effective material for delivering therapeutic HPV DNA vaccines,” please include data demonstrating the differences in cellular delivery efficiency among the various LNPs and EP used in the experiments. Results should be presented to show the differences in the number of DNA plasmid copies delivered into target tissue regions or specific cells under different delivery strategies, as well as the differences in the levels of expressed E6/E7 antigenic proteins. Of course, methods such as tissue fluorescence distribution or histochemistry are also very acceptable.

Reply: Thank you for your advice. The delivery DNA copies and E6/E7 protein expression are indeed important as you mentioned. But we only detected the HPV E7 expression in vitro, and did not do in vivo experiments and we will follow up with corresponding measurements.

Round 2

Reviewer 1 Report

Comments and Suggestions for Authors

Comments addressed.

Reviewer 2 Report

Comments and Suggestions for Authors The author's revised version of the manuscript has improved in quality, and most of the necessary experiments have been supplemented. I am okay with the current version.